# Stationary correlation pattern in highly non-stationary MEG recordings of healthy subjects and its relation to former EEG studies

**ArlexOscar Marín–García**[1⊙]**, J. Daniel Arzate-Mena**[2⊙]**, Mari Corsi-Cabrera**[3,4]**,
Zeidy Muñoz-Torres**[1,4]**, Paola Vanessa Olguín–Rodríguez**[1,2]**, Wady Aalexander Ríos–
Herrera**[4]**, AnaLeonor Rivera**[1,5]**, Markus F. Müller**[1,2,6]*

**1** Centro de Ciencias de la Complejidad, Universidad Nacional Autónoma de México, Ciudad de México,
México, **2** Centro de Investigación en Ciencias, Universidad Autónoma del Estado de Morelos, Cuernavaca,
Morelos, México, **3** Unidad de Neurodesarrollo, Instituto de Neurobiología, Universidad Nacional Autónoma
de México, Juriquilla, México, **4** Facultad de Psicología, Universidad Nacional Autónoma de México, Ciudad
de México, México, **5** Instituto de Ciencias Nucleares, Universidad Nacional Autónoma 15 de México, Ciudad
de México, México, **6** Centro Internacional de Ciencias A.C., Cuernavaca, Morelos, México

⊙ These authors contributed equally to this work.
* muellerm@uaem.mx

Sciences and Technology, PAKISTAN

**Data Availability Statement:** All relevant data is
presented within the paper and figures; however, if
necessary, the data of the correlation matrices can

## Abstract

In this study, we analyze magnetoencephalographic (MEG) recordings from 48 clinically
healthy subjects obtained from the Human Connectome Project (HCP) while they performed
a working memory task and a motor task. Our results reveal a well-developed, stable interre-
lation pattern that spans the entire scalp and is nearly universal, being almost task- and sub-
ject-independent. Additionally, we demonstrate that this pattern closely resembles a
stationary correlation pattern (SCP) observed in EEG signals under various physiological
and pathological conditions (the distribution of Pearson correlations are centered at about
0.75). Furthermore, we identify the most effective EEG reference for studying the brain's
functional network derived from lag-zero cross-correlations. We contextualize these findings
within the theory of complex dynamical systems operating near a critical point of a phase
transition.

## 1. Introduction

Human brain dynamics is highly nonstationary, as it is constantly exposed to ever changing
external stimuli and continuously controls internal regulatory circuits that are subject to con-
stant changes in physiological parameters. This nonstationary nature is reflected in drastic
morphological changes of empirical recordings like e.g., electroencephalographic recordings
(EEG) [1] as well as variations of statistical dependencies between signals, viz. the functional
network, as reported in [2–5], where a large reorganisation of the functional network was doc-
umented during the peri-ictal transition of focal onset seizures.

This apparent non-stationary nature of brain recordings suggests that bivariate measures
that can take both positive and negative values, such as the Pearson correlation, take both signs

be uploaded to a repository. The MEG data is open access and can be accessed at https://www.humanconnectome.org/study/hcp-young-adult. The information was extracted from the Reference Manual 1200 Subject Release from the Consortium Human Connectome Project, Page 189. Additional details, such as demographic status, are restricted. The sleep data for project No. FM/DI/028/2005, titled "Disturbances of alertness and controlled selective attention, caused by chronic insomnia, sleep deprivation, and diazepam" underwent review and approval by the Research and Ethics Commissions of the Psychology Faculty at the Universidad Nacional Autónoma de México (UNAM) during its ordinary session on December 4, 2005. All individuals provided their oral consent. However, due to ethical constraints and the sensitive nature of medical data collection, EEG data are available upon reasonable request after approval from the ethics committee. Requests and inquiries about the study can be directed to Dr. María Corsi Cabrera at corsi@unam.mx or Dr. Paz María Salazar Schettino, Head of the Research Division, at ciefm@unam.mx. It is important to note that the sleep EEG data have been utilized in other previously published studies, which can be found in Cabrera et al. 2012, DOI:https://doi.org/10.5665/sleep.1734, Olguín et al. 2018, DOI: https://doi.org/10.1016/j.neuroimage.2021.118763, and Rios-Herrera et al. 2019, DOI: https://doi.org/10.3389/fnins.2019.00941.

**Funding:** This work was supported by Consejo Nacional de Ciencia y Tecnología (CONACyT) (MF-M obtained the grant CF-263377 and AL-R CF-610285 ) and Dirección General de Asuntos del Personal Académico Programa de Apoyo a Proyectos de Investigación e Innovación Tecnológica de la Universidad Nacional Autónoma de México (DGAPA-PAPIIT, UNAM): AL-R is responsable of the Project IA208018 WA-R of the project IA100522, and Z-MT of the Project IN221324. The funders had no role in study design, data collection and analysis, decision to publish, or preparation of the manuscript.

**Competing interests:** The authors have declared that no competing interests exist.

with equal probability. Then, averages estimated via a running window approach over longer time intervals should tend rapidly to zero.

Counterintuitively, for EEG recordings, quite different results have been obtained. In [6, 7] EEG's have been measured monthly for groups of healthy women over periods of up to nine months. As a result, most of the correlation coefficients fluctuate negligibly around large average values, where averages have been taken not only over the different epochs but also over subjects. In [8] persistent connection patterns in electrocorticograms of 6 epilepsy patients have been found using graph theoretical techniques. These network templates emerge within time intervals as short as 100 seconds and stay stable during the long-term recordings. The authors suspect the existence of a metastable scaffold of brain connectivity, from which transient departures emerge continuously.

In the same spirit are the results presented in [9–11], where average zero-lag cross-correlation matrices not only show surprisingly high entries but also Pearson coefficients to estimate the similarity between two matrices took values of about 0.85. This outcome holds for the comparison of EEG segments covering different physiological states as well as the comparison between subjects. In [10] the EEG recordings of 31 healthy subjects that stem from three different laboratories have been considered: 10 subjects during the second night sleep, 11 elderly and 10 young subjects at rest during open and closed eyes, and in addition the peri-ictal transition of 9 patients suffering focal onset seizures. Not only the spatial correlation pattern averaged separately over e.g., sleep stages turn out to be almost indistinguishable. Pearson coefficients for the comparison of cross-correlation matrices averaged separately for different sleep stages of the same subject are systematically above 0.9, but also the average pattern observed before, during, and after an epileptic crisis reach on the average values above 0.85. Furthermore, the average correlation structure observed for all 40 subjects (31 healthy and 9 with epilepsy) is amazingly similar (the distribution of Pearson coefficients used as a similarity index between matrices are centred above 0.75). Note, this is true for averages over considerably long periods of at least one or two minutes. On shorter time scales, one may observe notable fluctuations (see e.g., Fig 6 of [10]). Also, for intracranial EEG recordings of epilepsy patients, an extremely stable linear cross-correlation pattern has been reported [4] The speed of convergence to the stationary pattern and its dependence on the window length is dealt with quantitatively below.

All the above-mentioned studies reporting this stationary correlation pattern (SCP) have been undertaken with EEG-recordings. One possible source of this phenomenon, inferred from EEG recordings, is volume conduction, where several (or all) electrodes are exposed to the electric field emanating from the same source [12, 13]). Volume conducted electrical activity that stems from a single source but is measured from two spatially separated electrodes has zero (or nearly zero) delay [14] but may cause a (positive or negative) phase shift of 0° as well as 180° in the cross-spectrum [15, 16]. Thus, instead of real, interconnected neuronal activity in distant regions of the scalp estimated by zero-lag cross-correlations, it is possible that only redundant information caused by volume conduction is measured.

Therefore, in [9], in addition to the zero-lag correlations, the weighted phase-lag index [16], which serves to eliminate the effects of volume conduction, and lagged cross-correlations, in which zero-lag coefficients were explicitly excluded, were estimated. In both cases, the previous results obtained with zero-lag cross-correlations were quantitatively confirmed. From this we conclude that volume conduction plays only a minor role in this respect, if any.

Another source of pronounced temporally stable interrelations between nearby or distant electrodes could be the chosen EEG-reference [17]. An EEG reference can produce a significant correlation offset, such that all electrodes become strongly correlated, or completely deform the functional network estimated by any synchronization measure [18, 19]. Such

stationary deformations can be so pronounced that the temporal variation of the true cross-correlation pattern generated by neuronal activity is totally overshadowed [17, 19]. Thus, another explanation for the unexpected observations of a well pronounced stationary cross-correlation pattern could be the chosen reference scheme. Therefore, an independent test seems necessary to determine whether such a stationary correlation pattern exists or whether it is just an artefact caused by the experimental design like e.g., the chosen EEG reference.

Ramos-Loyo et al. [20] reported a stationary correlation pattern in three groups of children (6,8, and 10 years old) during resting state and during odd-ball task in EEG recordings using the earlobe reference. The authors found high similarity between the correlation pattern across subjects (within or between the three groups) and conditions, expressed by Pearson coefficients above 0.8 for the comparison between subjects. However, the SCP reported in [20], is qualitatively different from that found in [9–11], when the median reference has been used. Regardless of the structural difference in the SCP when using earlobe or median reference, the EEG recordings contain a stable background of cross-correlations, generic in the sense that it is almost task and subject independent. Nonetheless, it remains unclear whether the stable structures observed in the aforementioned studies are solely a product of the EEG reference scheme, knowing that EEG references may induce strong linear interrelations extremely stable in time [17]. On the other hand, if there exists a genuine interrelation pattern covering the whole scalp, its true topological structure is unknown because the selected reference point may alter notably its topology [17, 19].

Furthermore, one may ask whether it is principally possible that a genuine spatially extended, pronounced cross-correlations, stable over extremely long time periods like a whole night recording, exists in an otherwise highly non-stationary complex system? Recall, nonstationarity means that the structure in phase space remains constant, viz. the attractor topology does not change [21]. The phase space is an abstract mathematical space whose dimension is equal to the number of degrees of freedom of a dynamical system. Each point in this space completely classifies a possible dynamic state of a system that describes a trajectory in this space during its temporal evolution. In other words, each point contains the minimal information that is necessary to obtain the maximal possible information about the dynamics of the underlying system.

Normally, a system does not traverse the entire phase space, but its dynamics are restricted to a spatially limited subset within this space, which is referred to as the invariant set or attractor. If parameters do not change (considerably), this structure remains (almost) constant.

In non-stationary systems, however, the topology of the attractor can change drastically, and in this case the relationship pattern within a multivariate record cannot be expected to remain constant (see Fig 6 of the supplementary of [10]). In this scenario, the observed stationary pattern would only be an artefact caused by the EEG reference.

Another source of nonstationary behaviour could be that a system is constantly exposed to external disturbances (stimuli) that lead to deviations from the invariant set in phase space, whose topological structure is stable in time. In this scenario, the system moves permanently on a transient trajectory in phase space, but always toward the stable invariant set. Then one would expect a stationary background pattern of interrelations, while dynamical aspects, like the response to an external stimulus, provoke permanent fluctuations around this stable pattern (see Fig 5 of the supplementary of [10]).

It remains evident that if one could distinguish between those scenarios one could reveal principal aspects of brain dynamics, although this seems impossible if one restricts solely on EEG recordings due to the problems delineated above.

MEG recordings, on the other hand, do not require a reference point and are therefore free of such deficiencies, since instead of the electrical potential, another physical quantity, the magnetic field response, is measured. Besides, MEG-recordings are less vulnerable to volume conduction (van den Broek et al. 1997). In particular, long-range correlations affect MEG recordings significantly less than EEG measurements [22, 23]. Moreover, in contrast to the EEG, the MEG is more sensitive to tangential sources from the sulci [24]. The relationship between functional connectivity in EEG and MEG has been found in several articles [18, 25] or power spectrum analysis [26] but not with the aim to identify and characterise a possible common stationary correlation pattern.

Thus, we hypothesise that by comparing average cross-correlation matrices estimated from MEG recordings, one can decide whether the specificities of the electroencephalographic measurements mimic stationary cross-correlations or whether the SCP is an expression of true permanent communication between even distant brain regions. Via a direct comparison of the obtained correlation matrices one might furthermore reveal the optimal EEG reference scheme for the construction of the functional network.

Thus, the main objectives of the present work are:

- to probe whether we detect a stationary interrelation pattern in MEG recordings during different physiological states, and thus rule out volume conduction as a possible source of this phenomenon, at least for long-range correlations.

- to quantify the similarity between the stable functional framework encountered in MEG-measurements and the framework revealed in EEG recordings. This comparison allows us to identify the optimal EEG reference scheme that causes less distortions of the functional network.

The accomplishment of these two objectives puts the above-mentioned findings on a more general footing. In the discussion section we place our results within established theoretical frameworks, arguing that they are congruent with the disproportionate energy consumption of the brain and fit the hypothesis that the brain is a complex system operating close to a critical point.

## 2. Methods

Here we use MEG-recordings of 48 subjects of the Human Connectome Project [27, 28]. From the total 1206 healthy young adults measured in this project we selected the MEG recordings of 48 participants under the criteria that subjects' recordings are present in all experimental conditions considered in this study. Subject information is collected in Table 1.

To investigate the relationships, we selected three physiological states. We considered the resting state, a state of the brain characterised by the occurrence of various functional networks that do not require active cognitive processes and activity during a working memory and motor task [27]. A detailed description of the experimental setup, processing pipelines, and quality controls are provided in [29, 30]; consult [31, 32] for an overview of the project.

### 2.1 Experimental design

**2.1.1 MEG data.**  MEG data is collected in approximately 3-hour sessions on a whole head MAGNES 3600 (4D Neuroimaging, San Diego, CA) at the Saint Louis University medical campus. All recordings were taken in a magnetically shielded room. The MEG system includes 248 magnetometer channels with a sampling rate of 2034.5101 Hz, 32bit/sample. Here we focus on 126 MEG channels that are common to all subjects. The data are then resampled to 508.63 Hz to reduce the size of the dataset.

**Table 1. Subject's information.** The information listed in Table 1 is extracted from the Reference Manual 1200 Subject Release from the Consortium Human Connectome Project Page 189. Gender and five year-age range are available as Open Access data. Additional information about e.g., demographic status is restricted. Corresponding recordings are available in https://www.humanconnectome.org/study/hcp-young-adult.

| Code | Sex | Age | Code | Sex | Age | Code | Sex | Age |
|---|---|---|---|---|---|---|---|---|
| 105923 | F | 31–35 | 191033 | F | 26–30 | 581450 | M | 22–25 |
| 106521 | F | 26–30 | 191437 | F | 31–35 | 599671 | M | 26–30 |
| 108323 | F | 26–30 | 192641 | F | 31–35 | 601127 | M | 22–25 |
| 109123 | M | 31–35 | 198653 | M | 22–25 | 660951 | F | 26–30 |
| 113922 | M | 31–35 | 204521 | F | 31–35 | 662551 | M | 26–30 |
| 116726 | M | 26–30 | 205119 | M | 31–35 | 667056 | M | 22–25 |
| 133019 | F | 26–30 | 212318 | F | 31–35 | 679770 | M | 26–30 |
| 140117 | F | 26–30 | 212823 | M | 22–25 | 680957 | F | 26–30 |
| 156334 | F | 26–30 | 255639 | F | 26–30 | 706040 | F | 22–25 |
| 162026 | F | 31–35 | 257845 | M | 26–30 | 707749 | M | 31–35 |
| 162935 | M | 22–25 | 283543 | M | 22–25 | 725751 | M | 26–30 |
| 164636 | M | 22–25 | 293748 | F | 31–35 | 735148 | M | 22–25 |
| 169040 | M | 22–25 | 353740 | M | 22–25 | 783462 | M | 22–25 |
| 175237 | F | 31–35 | 358144 | F | 26–30 | 814649 | M | 26–30 |
| 177746 | F | 26–30 | 406836 | F | 31–35 | 891667 | M | 26–30 |
| 185442 | M | 22–25 | 568963 | F | 31–35 | 898176 | M | 31–35 |

*Resting state (R).* Participants were instructed to relax with open eyes and to maintain fixation on a projected red crosshair on a dark background. This condition was recorded in three runs, each with 6 minutes duration.

*Working memory task (W1 and W2).* Consists of a 0-back and 2-back task. In two runs, pictures of tools or faces are presented in separate blocks in each run. Each run consists of 16 blocks, each with a 10-minute duration. 50% of the blocks within each run use a 2-back, the other half a 0-back working memory task. Participants are instructed to press a button in dependence of whether the presented picture matches the previous image (0-back) or the image presented three pictures before (2-back). Each picture is displayed for 2000ms. Thereafter, the subject must press a response button by using the index or middle finger of the right hand (if the image matches the target or not). The response must be given within a time-lapse of 500ms during which a fixation cross is presented in the centre of the screen. Here we consider 1.2 second segments, before and after the matching image appears. We treat both sets of segments separately, which are denoted by W1 and W2 in the sequel.

*Motor task (M1 and M2).* Consists also of two runs, each containing 32 blocks with a duration of 14-minutes. Two of the subjects' visual cues instructing the movement of either the right hand, left hand, right foot, or left foot are displayed. Each block of a movement type lasts 12 seconds (10 movements) and is preceded by a 3-second cue. The 32 blocks are divided into 16 hand movements (8 right and 8 left) and 16 foot movements (8 right and 8 left). In addition, there are nine 15-second fixation blocks per run. Again, we separate into 1.2 second segments just before and after the movement instruction is displayed. We refer to these two conditions as M1 and M2 in the sequel.

Datasets from the Human Connectome Project are selected to include only the datasets from subjects present in three different experimental conditions simultaneously: Resting state, Motor task, and Working memory task, this to provide a clear representation of a SCP for each subject in the dataset, which reduces the number of subjects to 48. As indicated in the preprocessing guidelines, time series are trimmed to 1.2s before and after the onset for movement instruction (Motor processing) and 1.2s before and 2.5s after the onset of a target image

(Working memory processing), whereas for the resting state condition time series are 4s long [29, 30]. To avoid including another source of variability into the analyses through data length differences, data segments for all conditions are set to 1.2s before and after a stimulus or instructions in the case of Motor and Working Memory processing, and 1.2s for Resting state.

**2.1.2 EEG data.** To attain the sleep recordings, 10 right-handed clinically healthy male subjects during night sleep were measured for two consecutive nights in the sleep laboratory of the Faculty of Psychology of the National Autonomous University of Mexico. The EEG data was recorded between 2005 and 2007 and was used for this retrospective study. We accessed this data for research purposes in 2016. The first night is used for adaptation to the experimental protocol and the second for EEG analysis. A 19 channel EEG was taken following the international 10–20 system using earlobe A1-electrode as a reference, with a recording frequency of 1024 Hz, which was then down-sampled to 128 Hz and band pass filtered between 1Hz to 40Hz. Eventually, data sets are re-referenced to other schemes to perform quantitative comparisons. A detailed description of the sleep EEG recordings can be found in [10]. The protocol was approved by the Ethical Committee of the Faculty of Medicine of the National Autonomous University of Mexico and followed the ethical standards of the Declaration of Helsinki (1964). All participants provided verbal informed consent. For Electroencephalographic data (EEG), M. C.C. had access to information that could identify individual participants during and after data collection.

## 2.2 Statistical analysis

The MEG data were filtered by a fourth-order Butterworth band pass filter between 1Hz and 30Hz. The filter borders are chosen to provide a direct comparison with the EEG data. Components below 1Hz are filtered out to avoid nonstationarities provoked by slow frequencies. Fast frequencies above 30 Hz, on the other hand, are eliminated as well because of the increasing influence of muscle activity at higher frequencies [33–35]. Electromyogenic artefacts have a broad frequency distribution reaching up to 200 Hz [34]. Given that the power spectra of genuine brain activity follows a power law ~$f^\beta$ with $\beta$ between 1 and 3 [36–38] the signal to noise ratio gets dramatically worse for higher frequency bands, which justifies our choice of the band pass borders [33–35]. Given that we consider a frequency range far below 100 Hz we did a downsampling to reduce the size of the data files. Note, due to the sampling theorem this does not cause a negative effect on our analysis.

For the filtering procedure we used a 4th-order butterworth filter in all cases. The choice of the filter order is a compromise to achieve well defined edges at the filter borders and to not artificially induce oscillations due to filtering, which might cause major distortions of cross-correlation estimates. Furthermore, it was shown that filtering in the time domain has several advantages in comparison to a Fourier filter [39].

For the task, MEGs pre-stimulus and post-stimulus segments of length 1.2 seconds are selected, to avoid introducing confounding factors, for the resting state condition also 1.2s segments have been extracted. In total, 20225 segments of the resting state, 17925 of the working memory tasks, and 39131 of the motor tasks have been used. Then, for each segment, the equal-time cross-correlation matrix is estimated and averaged separately for each experimental condition (resting state and the different task conditions) as well as for each subject.

Note, here we focus on linear interrelations within the multivariate dataset for two reasons: At first we aim to compare our results with those obtained in former studies, where also linear cross-correlations have been employed. Second, a large body of evidence has been collected over the last decades that linear cross-correlations cover most of the interactions between EEG channels [40, 41] and it seems that for nonlinear interrelations no stationary pattern can be

identified [9] Furthermore, even in the case of nonlinear dynamical systems in the chaotic regime, linear correlations perform equally well or even better than nonlinear measures when it comes to detecting possible coupling between signals [40, 42, 43]. However, we won't suggest that non-linear features are less important for the temporal development of a system just because they are less pronounced. The butterfly effect tells quite another story and the opportune detection and characterization of non-linear properties may be crucial for the understanding of the brain dynamics. But in this study we probe for the existence of a stationary linear interrelation pattern.

To test for topological similarity between matrices, the Pearson correlation coefficients between matrices are estimated. To this end, the non-diagonal elements of each matrix are ordered into a vector, which is subsequently normalised to zero mean and unit variance. Thereafter, the scalar product between those vectors provides an estimate for structural similarity. Note, the normalisation of the vectors ensures that this measure is not affected by the overall correlation strength. Solely the spatial distribution of positive and negative correlation coefficients influences the numerical outcome. Hence, in this context, one might understand the Pearson correlation coefficient as a topological similarity index for the comparison of two correlation matrices. To probe for the magnitude by which average cross-correlation structures are expressed, a property that is lost due to the before mentioned normalisation, we additionally apply the nonparametric Kolmogorov-Smirnov test. Besides estimating the similarity of average correlation matrices obtained in different experimental settings and different subjects, we also compared these matrices with those obtained for Iterated Amplitude Adjusted Fourier Transform surrogates (IAAFT, see [44]).

These surrogates conserve all linear properties of the signal like the power spectral density as well as the amplitude distribution in the time domain, but destroy all signatures of determinism as well as correlations between the different data channels of a multivariate dataset. This is achieved by replacing the Fourier phases by independent random numbers, uniformly distributed within $[0, 2\pi]$. Note, the Wiener-Khinchin theorem tells us that the Fourier transform of the auto-correlation function is the power spectral density of the signal. Thus, signatures of determinism, nonlinear behaviour as well as genuine interrelations between signals are encoded in the sequence of Fourier phases, or, a possible correlation between amplitudes and phases [45–47]. Therefore, these surrogates represent noise sharing the same linear univariate properties as the original recordings.

At this point it is crucial to conserve the power spectral densities of the signals, given that the amount of random cross-correlations [3, 48–51] depend strongly on the spectral content of the signals [4]. In particular, the amount of random cross-correlations increase drastically with a notable increase of slow wave components, like in deep sleep periods or during the immediate post-seizure period [4, 10]. Thus, these surrogates adequately represent the null hypothesis of zero genuine cross-correlations and a bootstrap method allows for estimating appropriately the significance values of the numerical results.

To compare the 126 magnetometer channels of the MEG with the standard 19 channel scalp EEG of the international 10–20 system, we proceed in three different ways. On the one hand, signals of the MEG-detectors closest to the 10–20 electrodes are chosen to construct the zero-lag cross-correlation matrix (panel **a** of the cartoon shown in Fig 1). This procedure is denoted by configuration 1 (C1). In a second procedure (C2), we estimate the average of the signals recorded by the MEG-detectors closest to the 10/20 positions (panel **b** of Fig 1). Then we construct the corresponding cross-correlation matrix of the average signals. Finally, we also compute zero-lag cross-correlation coefficients between all possible pairs of neighbours belonging to two different 10/20 locations of the EEG (C3).

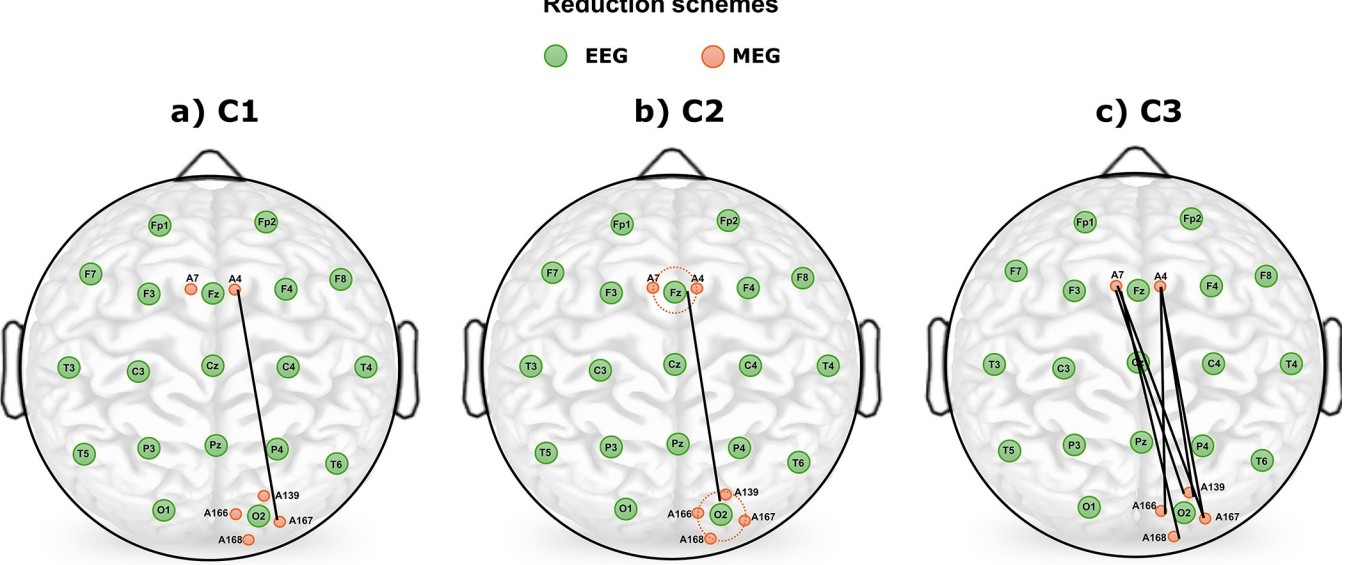

**Fig 1. Illustration of the different reduction schemes applied to MEG signals to simulate the international 10–20 system of EEG electrodes.** Green circles indicate the positions of EEG-electrodes, and red circles closest MEG-detectors of EEG channels Fz and O2. Panel **a**, **b**, and **c** refer to connections scheme C1, C2, and C3 respectively. A detailed explanation is provided in the method section of the main text.

Thereafter, the average over those correlation coefficients is taken as an estimate (panel **c** of Fig 1). For example, for the electrode F4 the A58 MEG position is chosen as the closest detector, and the MEG channels A86, A35, A58, A85, A34, A57, A84 as the closest neighbours. The complete scheme of the assignment between MEG detectors and EEG-electrode positions is listed in Table 2.

Considering that the spatial resolution of EEG recordings is about 10 times worse than that of the MEG, we find it mandatory to probe for different reduction schemes. While we expect that configurations C2 and C3 provide quantitatively similar results, estimates of scheme C1 might be biased from the specific location of the MEG detector. Furthermore, we hypothesise that the correlation pattern derived with configurations C2 and C3 may be closest to the SCP observed in EEG recordings, as the two different averaging schemes should simulate the poorer spatial resolution of the EEG.

A graphical representation of the entire pre-processing pipeline can be found in Fig 2.

## 3. Results

The presence of pronounced stationary correlation structures in otherwise highly non-stationary empirical recordings is such a surprising observation, providing relevant information about the basic principles of the underlying dynamics. Therefore, in the first step, we investigate whether we can identify a stationary cross-correlation pattern (SCP) in MEG data that is almost independent of the physiological state of a subject, i.e., that is almost task-independent (M1, M2, W1, W2, and R).

In Fig 3A–3F one gains a visual impression of the similarity of the average cross-correlation matrices obtained for a subject in different task conditions. Hereby, we averaged the correlation matrices of all trials separately for the two motor task segments M1 and M2 (panels **a** and **b**), for the working memory task segments W1 and W2 (panels **c** and **d**), as well as for the resting state R (panel **e**). Just by eye revision, it remains clear that the average matrices obtained for the different conditions are strikingly similar. Although the magnitude of the (anti)

**Table 2. MEG scheme reduction to the 10–20 EEG system.** MEG detectors selected for the simulation of the international 10/20 EEG system. The first column corresponds to EEG-electrodes, the second column corresponds to C1 configuration of MEG sensors. The third column denotes the MEG sensors used in schemes C2 and C3.

| 10–20 System | Configuration C1 | Configuration C2 & C3 |
|---|---|---|
| Fp1 | A91 | A92, A91 |
| F3 | A40 | A64, A39, A21, A22, A40, A65, A66 |
| C3 | A44 | A23, A24, A25, A26, A42, A43, A44, A67, A69 |
| P3 | A72 | A46, A47, A71, A72, A100, A101, A103 |
| F7 | A95 | A95, A127, A93, A125, A154 |
| T3 | A130 | A97, A98, A99, A130, A131, 158, A96, A156 |
| T5 | A160 | A132, A133, A134, A181, A160, A161 |
| O1 | A184 | A135, A163, A184 |
| Fz | A4 | A7, A4 |
| Cz | A10 and A14 | A3, A9, A10, A14, A16 |
| Pz | A49 | A28, A29, A48, A49, A50, A74, A75, A76 |
| Fp2 | A151 | A151 |
| F4 | A58 | A86, A35, A58, A85, A34, A57, A84 |
| C4 | A54 | A30, A31, A32, A33, A53, A54, A55, A56, A80, A82, A83 |
| P4 | A78 | A51, A77, A78, A79, A107, A108, A109, A110 |
| F8 | A115 and A116 | A117, A116, A115 |
| T4 | A144 | A113, A114, A144, A143, A171 |
| T6 | A169 | A140, A141, A142, A170, A168, A169, A191 |
| 02 | A167 | A139, A166, A167, A168 |

correlations derived for the motor tasks seems somewhat reduced, the spatial structure between all matrices shows only marginal changes, such that even an average over all conditions, but separately for each subject (panel **f**), seems justified.

This situation is quantified in the following panels of Fig 3. Panel **g** shows separately for each subject the colour-coded Pearson coefficients obtained for the pairwise comparison of the average matrices obtained for the 5 task conditions. The highest scores are obtained for the set of segments obtained by the same task (e.g., M1 and M2, or W1 and W2). For these comparisons, the scores are only slightly below one. In all cases, however, the similarity index is above 0.92, indicating extremely high topological similarity.

A similar result is obtained when we compare the average cross-correlation matrices of different subjects separately for each condition. Panel **h** of Fig 3 shows the cumulative probability distributions of the corresponding Pearson coefficients. Lowest values are obtained for the resting state and the two motor tasks, which might be due to muscle artefacts that contaminate to a certain degree the empirical recordings. However, the numerical estimates never fall below 0.83, which we still consider a remarkably high value. Finally, when averaging over all conditions, one obtains one correlation matrix for each subject, the so-called stationary correlation pattern (SCP). In panel **i** of Fig 3 we show the cumulative probability distribution for the inter-subject comparisons of these overall averaged correlation patterns. Again, we obtain strikingly high values above 0.87 for all cases.

In order to support and validate the results presented, we computed a post-hoc power two tails test for Pearson correlation coefficient, which revealed a power value of 0.994 and a p value of 0.001 considering a sample size of 48 subjects and a correlation value of 0.7, a smaller value than the observed in the correlation analyses.

We also probe for the significance of these results by comparison with IAAFT-surrogates. To this end, we generated for each of the 5 conditions separately average matrices based on

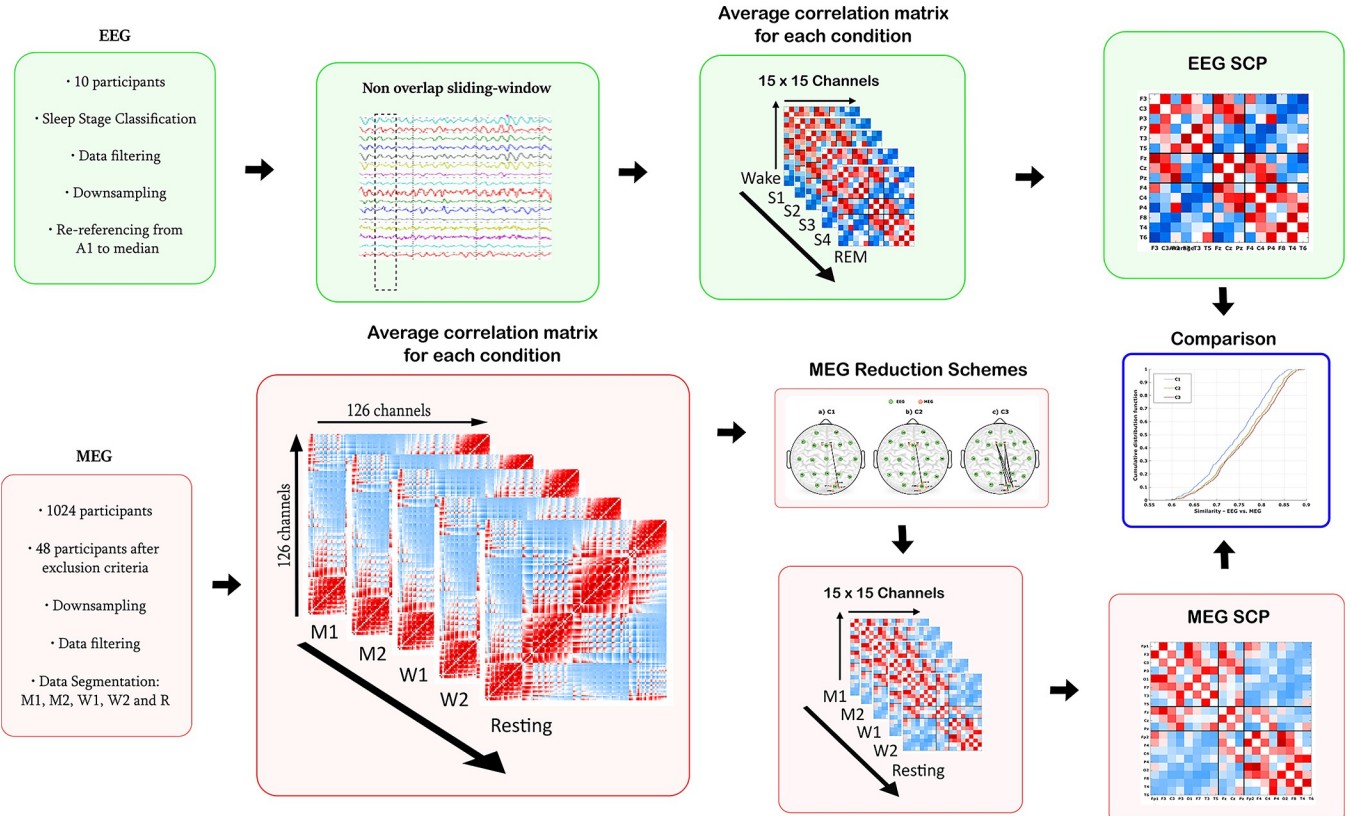

**Fig 2. Illustration of the methodology applied in this study, namely the experimental data and the data processing pipelines.** A detailed description can be found in the main text.

those surrogate data, that share the same linear univariate properties as the original data (namely the same power spectrum, viz. autocorrelation function, as well as the same amplitude distribution in the time domain) but lack any kind of genuine cross-correlation between the data channels. The matrix elements obtained for the surrogates exclusively display random cross-correlations [3, 48–51]. A Kolmogorov Smirnov reveals p-values far below $10^{-6}$. Thus, even when applying a Bonferroni correction for multiple testing of the 5 conditions (which provides a factor of 10 for the correction), the significance value of our results is well below the 1% level. Hence, also for MEG-recordings we obtain a generic genuine linear interrelation pattern in the sense that it does not alter drastically neither between different physiological states nor across subjects. The question remains whether this cross-correlation structure is in any way related to previous findings from the analysis of EEG signals. This is a relevant but non-trivial question because (i) the correlation patterns obtained from MEG recordings are not disturbed by any reference point as it is the case for the EEG [17, 19] and (ii) in MEG one not only measures a different physical quantity than in EEG recordings, but also the neuronal populations generating these signals do not necessarily coincide. While the EEG signal is largely fed by neuronal dipoles in the gyri, the MEG is mainly generated by the magnetic field associated with the dipoles of neurons located in the sulcus [24, 52]. A comparison of the average correlation structure resulting from both types of signals is shown for two subjects in Fig 4.

In the first row we show the interrelation matrices of a representative subject of the MEG-experiment derived for the reduction schemes outlined above. As one notices by visual inspection, the spatial structure of the resulting interrelation matrices is quite similar. A quantitative

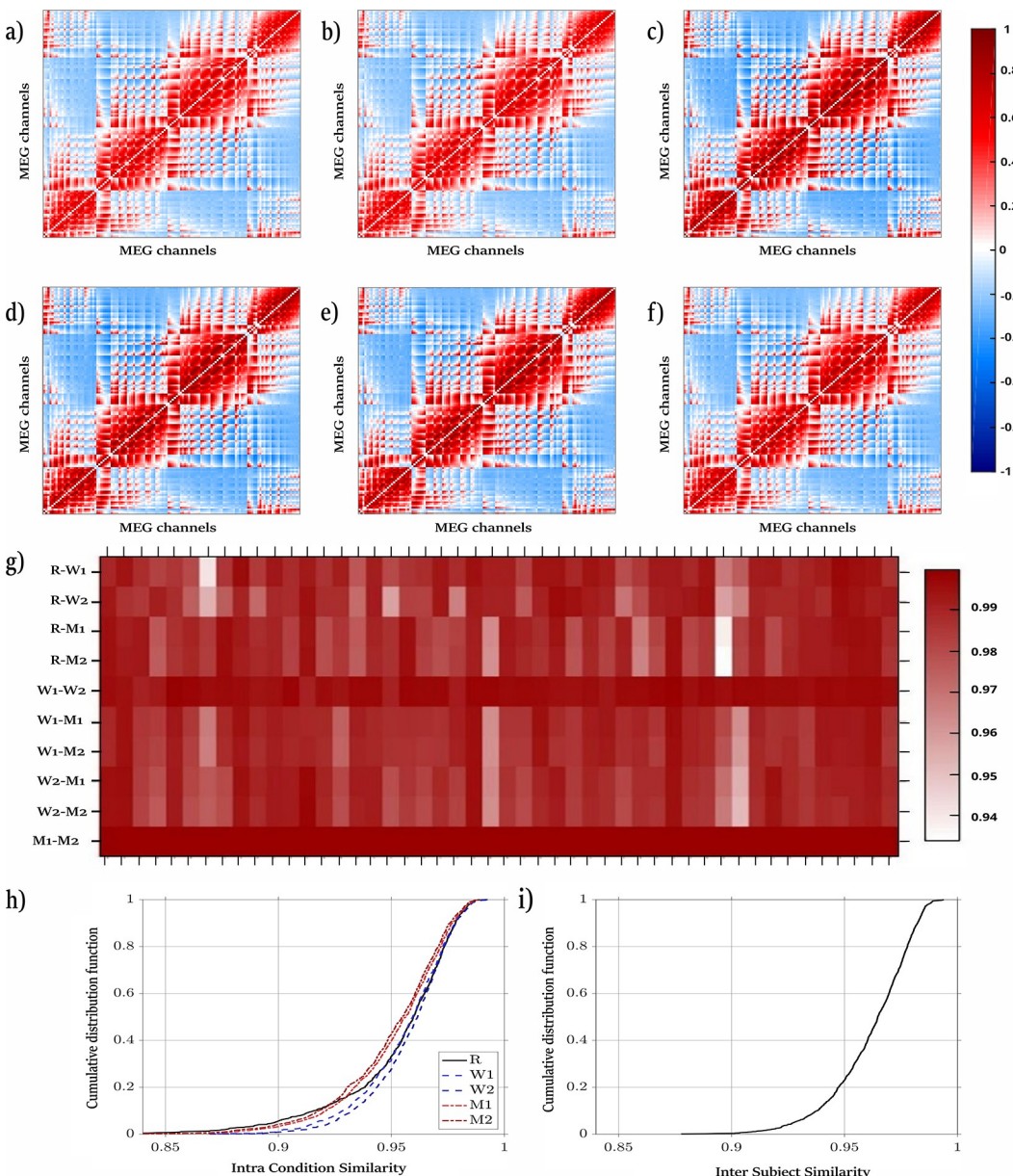

**Fig 3. Stationary correlation pattern (SCP) in MEG recordings.** Panels **a** to **f** show average cross-correlation matrices derived from the 126 MEG detectors of a subject with ID number 105923. Averages are taken separately for different task conditions: **a** and **b** correspond to the motor task M1 and M2, **c** and **d** to the working memory task W1 and W2, **e** for the resting state R and **f** displays the cross-correlation matrix averaged over all trials of the 5 conditions. Panel **g** shows for each subject the topological similarity between correlation matrices averaged separately for each condition. Panel **h** displays the empirical cumulative probability distribution of the topological similarity between subjects separately for each condition. Finally, panel **i** shows the same for the correlation matrices averaged over all 5 conditions (SCP) separately for each subject.

comparison between the three reduction schemes of the stationary MEG pattern obtained for all subjects confirms this visual impression. Pearson's coefficients are all above 0.96, where the closest similarity was found between scheme C2 and C3 (corresponding probability distributions are not shown).

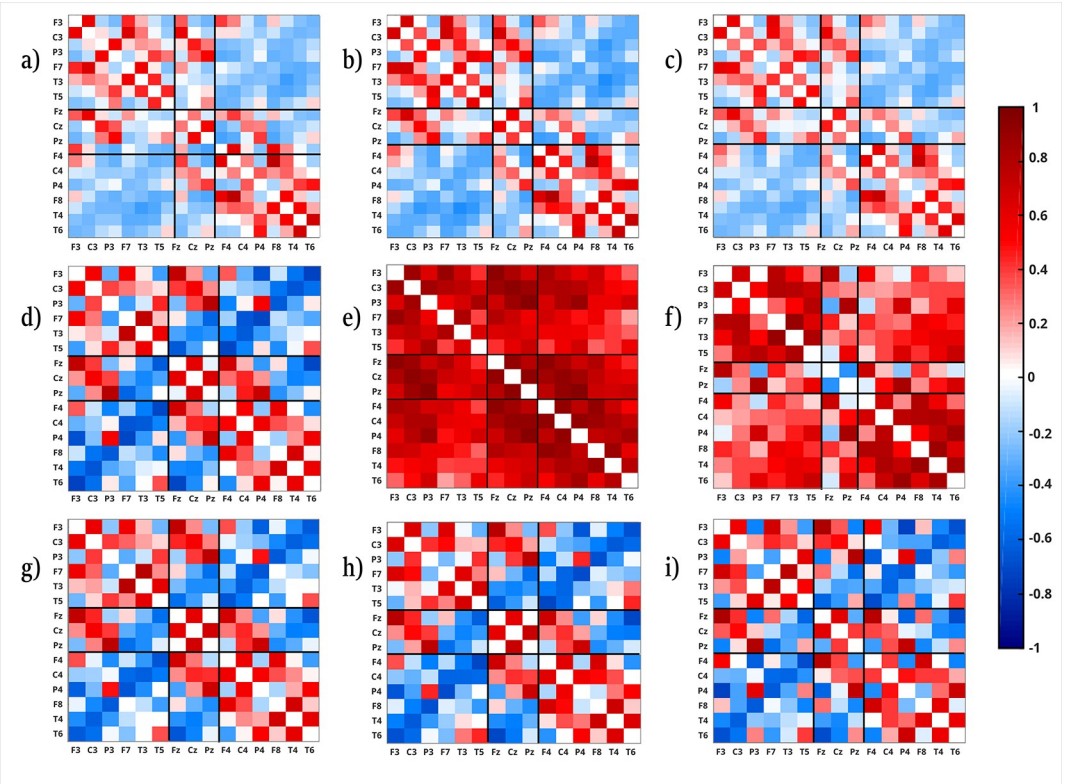

**Fig 4. Qualitative comparison of MEG and EEG correlation matrices.** Average correlation matrices derived from MEG and EEG recordings. The upper row shows the reduced correlation matrices of MEG-recordings averaged over all 5 physiological conditions of the subject with ID number 105923. Reduction scheme C1 to C3 is used for panel **a** to **c** respectively. In the second row, cross-correlation matrices derived from EEG recordings using different reference schemes, averaged over sleep stage 2 of a whole night sleep of a clinical healthy subject (subject 10 of [10]) is shown. For panel **d** the global average has been chosen, for matrix **e** the earlobe electrode A1 has been taken and, finally, for matrix **f** the active electrode Cz is used as the EEG-reference. The last row displays cross-correlation matrices derived from the EEG recording of the same subject during night sleep, using the median instead of the global average as a reference scheme. From panel **g** to **i** correlation pattern is averaged over sleep stage 2, slow wave sleep, and REM sleep respectively. The order of the electrodes (from left to the right, from below to above) in each of the matrices is: Fp1, F3, C3, P3, O1, F7, T3, T5, Fz, Cz, Pz, Fp2, F4, C4, P4, O2, F8, T4, T6.

The next row displays mean cross-correlation pattern derived from EEG signals of subject 10 of [10] during sleep stage 2 in three different reference schemes: global average (panel **d**), earlobe A1 (panel **e**) and the average interrelation pattern using the active electrode Cz as reference (panel **f**). For the case of these subjects, we notice the strongest similarity between the interrelation matrices derived from MEG and EEG for the global average reference. As already reported in [19], reference A1-A2 induces redundant information on all active EEG channels. This significantly increases the overall correlation structure towards positive values. On the other hand, the usage of an active electrode (like in bipolar schemes) provokes drastic distortions of the spatial correlation structure (see [19] for a detailed discussion).

Finally, in panels **g** to **i**, mean cross-correlation matrices using the median reference, the non-parametric version of the global average, are shown for sleep stage 2, slow wave sleep and rapid eye movement sleep (REM) respectively. One notices the striking similarity between the matrix derived for the median reference, and the global average, as well as the reduced MEG-matrices. In [19], the median reference is recommended based on model calculations because it does not depend on outliers and preserves nicely the actual spatial interrelation structure of the functional brain network. However, these results are based on data derived from

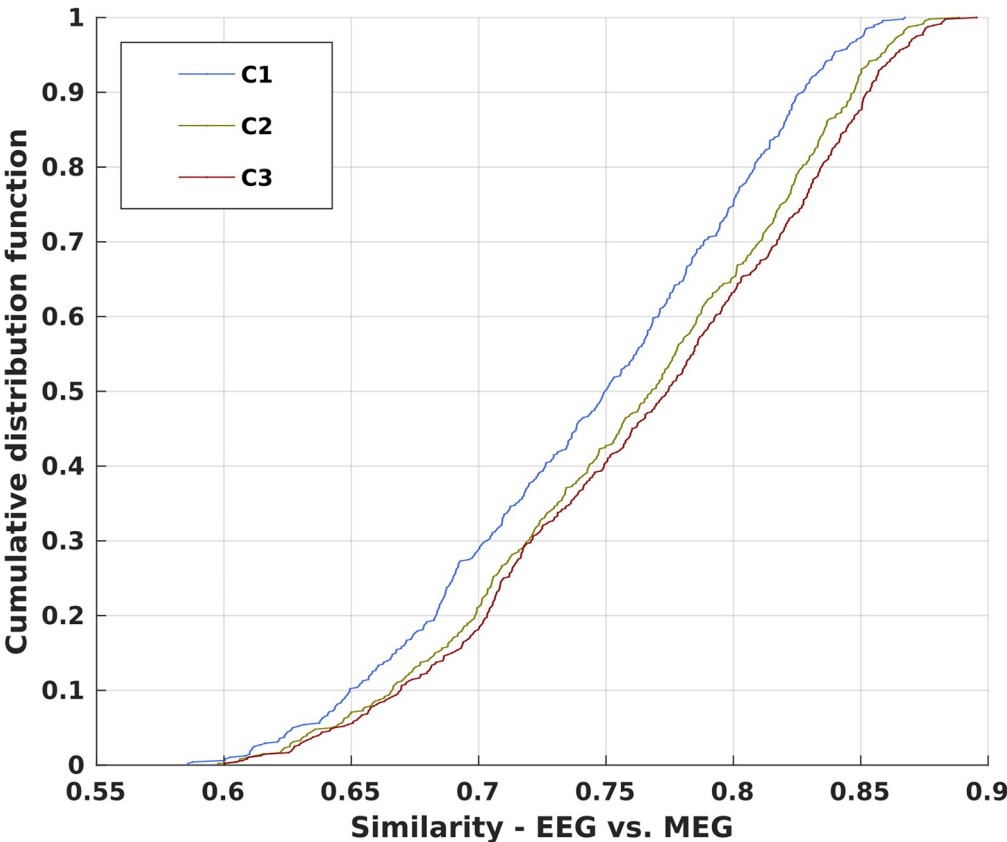

**Fig 5. Similarity of the stationary correlation pattern (SCP) between MEG and EEG recordings.** Pairwise comparison of the stationary correlation patterns obtained for the sleep EEG's of the 10 clinically healthy subjects using the median reference and the reduced average correlation matrices obtained from the MEG recordings of the 48 subjects. The cumulative probability distributions of the Pearson coefficients are shown for the separate comparisons with the three different reduction schemes C1 to C3.

theoretical models; instead, here we show a more objective comparison with real MEG recordings. Fig 4 displays only the comparison of two arbitrarily chosen subjects. A statistically more relevant picture is drawn in Fig 5.

Fig 5 displays the cumulative probability distributions of the similarity index between matrices (Pearson coefficient) to quantify the topological similarity between the stationary correlation pattern (SCP) of the 10 sleep EEG's (using the median reference) and the SCP of all 48 subjects for the three reduction schemes of the MEG recordings. Lowest values are about 0.6, but more than 50% of the Pearson coefficients encounter values above 0.75, a considerably high value. Each of the three probability distributions has approximately the same width, but the distribution obtained for comparison with reduction scheme C1, where the closest MEG sensor was used for reducing the number of data channels, is slightly shifted to lower values. This was to be expected, as the possible peculiarities of recording a single channel may affect these statistics, while each of the two averaging processes of schemes C2 and C3 provides a more representative signal of brain activity around the location of an EEG electrode, which has a reduced spatial resolution than MEG detectors. However, regardless of the reduction scheme, we find a high degree of similarity between the averaged EEG patterns obtained for the median reference and those obtained for MEG recordings.

Given the high similarity between the average correlation matrices of the subjects in both groups, we estimated an average correlation matrix for the MEG group for each of the three

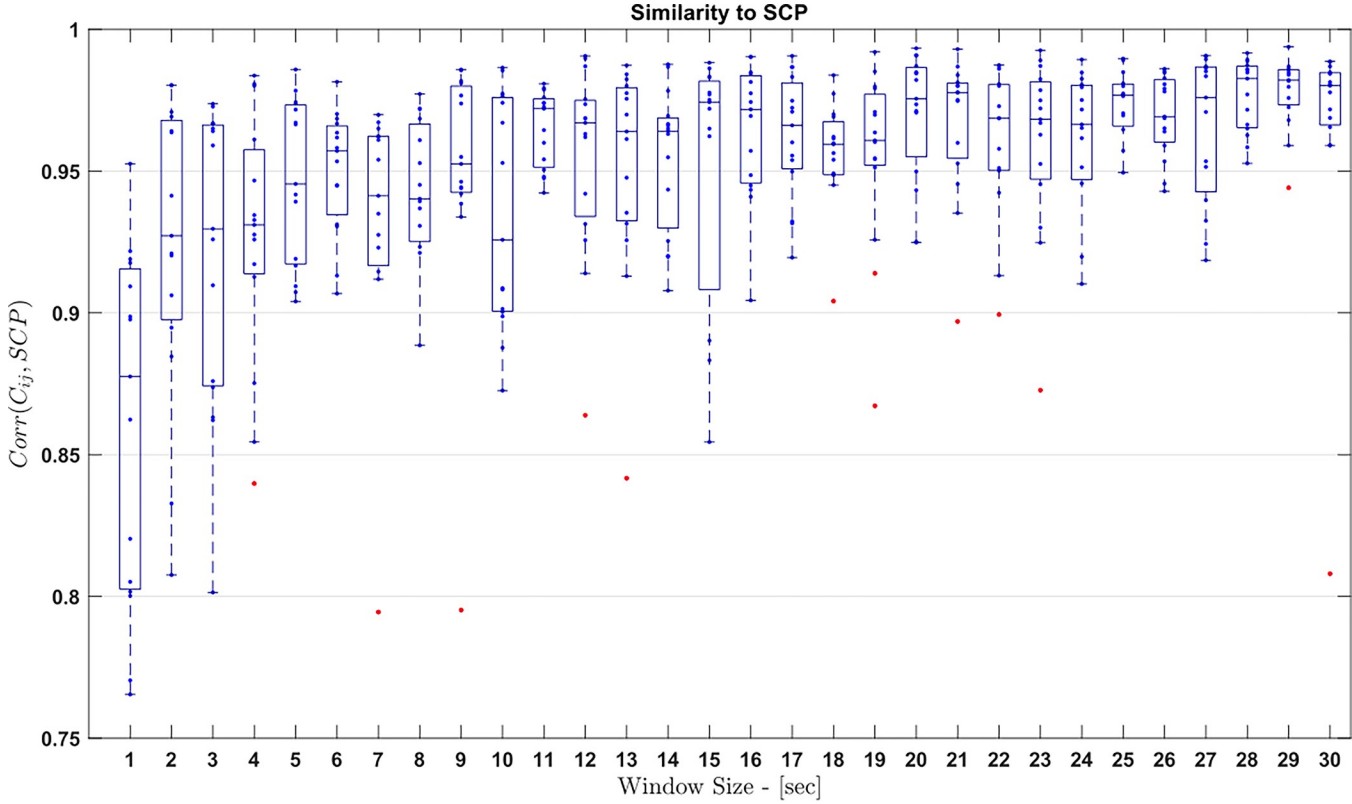

**Fig 6. Similarity of the stationary correlation pattern (SCP) and correlation matrices estimated for data windows of varying sizes.** For each length we chose randomly 15 windows of a given length. The samples of the resulting Pearson coefficients are drawn as box plots as a function of the window sizes.

reduction schemes as well as for the EEG group of healthy subjects during sleep. We estimated the Pearson coefficient for evidence of topological similarity and obtained 0.80, 0.81 and 0.82 for the three reduction schemes.

Finally, we aim to estimate the time scales on which the average matrices converge to the stationary pattern. For this purpose, we calculate the similarity between the correlation matrix estimated over a time window of a certain length and the stationary pattern. Based on the sleep recordings, we selected 15 randomly chosen data windows for each window length. The distribution of the corresponding Pearson coefficients of these samples is shown as boxplots in Fig 6.

Fig 6 shows that the fluctuations decrease rapidly with the window length and that even with small window sizes of only one second, the Pearson coefficients reach comparatively large values above 0.75. For segment lengths above 10 seconds average values as well as the width of the distribution saturate somehow. That implies that the time scale for the convergence to the stationary pattern is about 150 seconds for extracranial recordings. This estimate corresponds approximately to the scale of 100 seconds determined for intracranial measurements [8].

## 4. Discussion

In the present study, we pursued several main objectives. First, we wanted to verify whether a stationary, well-pronounced correlation pattern is also evident in MEG signals. Fig 3 provides abundant evidence in favour of this hypothesis. By comparing average cross-correlation matrices for 5 experimental conditions (R, W1, W2, M1 and M2) we show that the average patterns are practically independent of the physiological state. Furthermore, we show that these

patterns are, like in the case of EEG recordings [9–11], generic in the sense that they are almost subject independent. Comparison of the overall average of 48 different subjects provides Pearson coefficients above 0.87 (panel **i** of Fig 3), a strikingly high value. A Kolmogorov-Smirnov test was carried in contrast to IAAFT-surrogates and confirms that these findings are highly significant with p-values below $10^{-6}$.

Second, we wanted to test whether this temporally stable pattern corresponds to that found in EEG signals in previous studies, thereby revealing which EEG reference leads to the greatest similarities and, thus, searching for the reference scheme that causes less distortions of the functional network. Corresponding results are summarised in Figs 4 and 5.

We provide clear evidence that the stationary pattern found in EEG's is remarkably alike to that estimated in the MEG signals (Fig 5). When looking at these results, it should be noted that MEG signals not only measure a different physical quantity that comes from different neuronal populations than those captured by EEG recordings, but also that the physiological conditions of the two measurements were completely different. While the EEG's were recorded during night sleep, i.e. show abrupt and drastic morphological changes during the sleep cycles, the MEG's were measured while the subjects were awake and had to solve a cognitive and motor task. Nevertheless, Pearson coefficients comparing the average cross-correlation pattern of both measurements range between 0.6 to 0.9, which clearly proves the generic nature of this stationary interrelation structure. The fact that we additionally used three different reduction schemes to fit the multivariate MEG recordings to the international 10–20 system of the EEG underlines the robustness of these results.

Furthermore, we confirmed the outcome of extensive model calculations reported in [19], but here dealing with real world signals. Since the MEG does not require a reference point like the measurement of the electrical potential, it provides an ideal framework for comparison. Hence, the stationary pattern is not artificially generated by an EEG-reference, which could also produce timely stable spurious correlations [17, 19], but is certainly a reflection of permanent, well-coordinated brain activity.

We can further state that the high similarity shown in Figs 4 and 5 could only be achieved if the median (or alternatively the global average with slightly worse results) is used [19]. Other reference schemes cause relevant distortions of the correlation pattern and are therefore not suitable for the study of the functional network of the brain dynamics. Thus, the use of the median reference leads on average to almost the same (reference-free) correlation pattern obtained for MEG recordings, leading to the conclusion that this reference scheme is most favourable when studying the functional network of brain activity derived by zero-lag cross-correlations.

Volume conduction may alter the real part of the cross-spectra between two signals, recorded from spatially distant detectors [15, 53] by a positive or negative shift, causing phase differences of zero or 180˚ respectively [15, 16]. In [9] lagged cross-correlations as well as the weighted phase lag index [16], which is a modified version of the imaginary part of the coherency [53], has been used for the estimation of mean interrelation pattern. Note, both estimators explicitly exclude volume conduction effects. The authors yield numerical results that are like those estimated by the zero-lag cross correlations.

Considering that MEG recordings are much less affected by volume conduction [54], which is particularly true for long-range interrelations [22, 23], we can also exclude such trivial effects as the cause of our observations. The high similarity between the patterns obtained for the MEG and EEG recordings underline these results emphatically.

Strong evidence that the stationary cross-correlation pattern is a universal feature of brain dynamics was also presented in [11] via a simultaneous EEG-fMRI study. There, fluctuations around the stationary EEG-pattern have been used to predict fMRI resting state networks,

yielding specificity values above 98%. The authors conclude that the well established concept of large-scale fMRI resting state networks and the stationary pattern observed in extracranial EEG recordings are two different expressions of the same phenomenon–the ongoing neural activity. It is to say, even at rest the brain is permanently active [55–59], producing the largest possible set of different spatio-temporal structures of synchronous neural activity [60–65] expressed by power laws [66, 67]. Note, probability distributions following a power law do not have a characteristic scale (their average value diverges), and, thus, there is no maximum/minimum scale that may occur, but all possible scales appear with a finite probability. Therefore, if the spatial scales or the live times of synchronisation pattern follow a power law, all possible sizes are permanently present, which implies the maximal possible variability. That explains also the disproportionate high energy consumption of the brain, even in resting state conditions [55, 57, 58, 68].

According to these and previously published results, it appears that the activity generating this stable scaffold of distinct interactions between distant brain regions represents a kind of basal dynamic state. Any type of external perturbation, or spontaneous internal activity distinct from rest, provoke deviations from this background pattern. In the theory of dynamical systems such behaviour is known as "attractor activity", is to say, a certain set of dynamical states is "attractive" for the system in the sense that it returns rapidly to this preferred set after provoked deviations. This behaviour is invariant to the type (and partially to the strength) of the external perturbations (viz. external stimuli), the reason why the set forming this attractor in state space is also called the "invariant set". It seems that this well coordinated ongoing activity optimises brain dynamics in the sense that the system is maintained in an optimal mode of information processing. Thus, the high cost of the unproportional energy consumption for the maintenance of the pronounced cross-correlation pattern, is compensated by a most efficient operational mode, such that the additional energy cost of cognitive processes or the reaction of external stimuli is tiny [57, 58, 68].

Ongoing activity decoded in EEG-signals is also interpreted as the organised sequence of so-called microstates [69], which are specific spatial patterns of the EEG-potential covering the whole scalp. Although such pattern is defined by a univariate quantity, namely the signal power, and they are short lasting events of about 60-150ms with transition periods of only 25-50ms, there might be a link between the rapid alternations of the microstates and the stationary pattern, viz. a permanent interrelation structure stable over large time periods.

At first, also the sequence of microstates is strongly related to fMRI resting state networks, as it could be revealed by simultaneous EEG-fMRI experiments [70–72], although the findings of the three studies vary [73]. Second, the phenomenon of microstates is also related to criticality, given the scale free characteristics of microstate sequences [74] and third, they are interpreted as the "basic building blocks of consciousness" and "atoms of thoughts". Hence, they are interpreted as chunks of higher order cognitive processes, although the physiological interpretation of the microstates in terms of cognitive processes is quite dispersed (see e.g., [75]). In [10], it was hypothesised that such transient dynamics, like the response to an external stimulus, cognitive processes, and different mental states, are translated as fluctuations around the stationary correlation pattern. In this spirit, the sequence of microstates can be interpreted as these fluctuating patterns, such hypothesis is in concordance with the results obtained in the EEG-fMRI studies [11, 70–72].

In any case, these rich patterns of ongoing activity cannot be completely uncoordinated. Temporarily stable, long-range correlations must ensure that no "undesirable" patterns emerge which could, for example, set in motion unsuitable motor activity. This explains the presence of the well pronounced stationary spatial correlation pattern, visible in extracranial MEG as well as EEG recordings, that cover the whole scalp region [9, 10]. Thus, for the execution of a

given cognitive or motor task, it is not necessary to generate the corresponding neuronal activity pattern from scratch, but only to modulate the existing spontaneous activity appropriately. This behaviour increases the efficiency of the neural network's information processing and accounts for the small increase in energy consumption during task execution [56]. This has been identified as the dynamics near a critical point of a phase transition, where activity and rest, excitability and inhibition, synchronization and desynchronization are in a fine-tuned balance (see [61] and references therein). Note, the existence of the pronounced spatially extended cross-correlation pattern is not a proof of criticality, but it is consistent with the criticality hypothesis.

In summary, the present study provides further evidence of the universal character of a well pronounced, large-scale cross-correlation pattern, a point of view that opens new perspectives of brain research, because non-stationary dynamical features, like task related actions, should be encoded in specific deviations from (or fluctuations around) this stationary correlation pattern [10, 11]. Furthermore, we show that the median reference is most suitable when it is aimed to construct and analyse the functional network of the brain activity.

## Author Contributions

**Conceptualization:** ArlexOscar Marín–García, J. Daniel Arzate-Mena, Markus F. Müller.

**Data curation:** ArlexOscar Marín–García, J. Daniel Arzate-Mena, Mari Corsi-Cabrera, Paola Vanessa Olguín–Rodríguez.

**Formal analysis:** ArlexOscar Marín–García, J. Daniel Arzate-Mena, Paola Vanessa Olguín–Rodríguez, Markus F. Müller.

**Funding acquisition:** Zeidy Muñoz-Torres, Wady Aalexander Ríos–Herrera, AnaLeonor Rivera, Markus F. Müller.

**Investigation:** ArlexOscar Marín–García.

**Methodology:** ArlexOscar Marín–García, J. Daniel Arzate-Mena, Paola Vanessa Olguín–Rodríguez, Markus F. Müller.

**Project administration:** Markus F. Müller.

**Resources:** Mari Corsi-Cabrera, Markus F. Müller.

**Software:** ArlexOscar Marín–García, J. Daniel Arzate-Mena, Paola Vanessa Olguín–Rodríguez.

**Supervision:** Markus F. Müller.

**Validation:** J. Daniel Arzate-Mena, Paola Vanessa Olguín–Rodríguez, Markus F. Müller.

**Visualization:** ArlexOscar Marín–García, J. Daniel Arzate-Mena, Paola Vanessa Olguín–Rodríguez.

**Writing – original draft:** Markus F. Müller.

**Writing – review & editing:** ArlexOscar Marín–García, J. Daniel Arzate-Mena, Mari Corsi-Cabrera, Zeidy Muñoz-Torres, Paola Vanessa Olguín–Rodríguez, Wady Aalexander Ríos–Herrera, AnaLeonor Rivera, Markus F. Müller.

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
