## [Decision Letter · Decision Letter 0]

21 Feb 2024

PONE-D-23-40266Stationary Correlation Pattern in highly non-stationary MEG recordings of healthy subjects and its relation to former EEG studiesPLOS ONE

Dear Dr. Müller,

Thank you for submitting your manuscript to PLOS ONE. After careful consideration, we feel that it has merit but does not fully meet PLOS ONE’s publication criteria as it currently stands. Therefore, we invite you to submit a revised version of the manuscript that addresses the points raised during the review process.

We look forward to receiving your revised manuscript.

Kind regards,

Umer Asgher, PhD

Academic Editor

PLOS ONE

Journal Requirements:

4. Please expand the acronym “CoNaCyT” (as indicated in your financial disclosure) so that it states the name of your funders in full.

5. Please amend your manuscript to include your abstract after the title page.

**Additional Editor Comments:**

Based on reviewer's feedback, major revision is recommended.

Reviewers' comments:

Reviewer's Responses to Questions

**Comments to the Author**

1. Is the manuscript technically sound, and do the data support the conclusions?

Reviewer #1: No

Reviewer #2: Partly

2. Has the statistical analysis been performed appropriately and rigorously? 

Reviewer #1: No

Reviewer #2: Yes

3. Have the authors made all data underlying the findings in their manuscript fully available?

Reviewer #1: Yes

Reviewer #2: No

4. Is the manuscript presented in an intelligible fashion and written in standard English?

Reviewer #1: Yes

Reviewer #2: Yes

5. Review Comments to the Author

Reviewer #1: I have reviewed the manuscript entitled "Stationary Correlation Pattern in highly non-stationary MEG recordings of healthy subjects and its relation to former EEG studies". It is interesting and has some minor novelty but is not accepted until the authors address the following comments:

- The title includes "stationary correlation", but the manuscript did not include rigorous definitions nor implementation of stationarity tests.

- Abstract is not well written. It is too convoluted and not clear. What do you mean by stable structures in "stable structures in otherwise highly nonstationary multivariate recordings"?

- Authors write that they analyzed Magnetoencephalographic recordings (MEG) of 48 clinically healthy subjects obtained by the Human Connectome Project (HCP). However, in the literature review, many of the examples were based on signals with neurological disorders. Please explain the level of accuracy in this literature review?

- In the Introduction, authors write "over longer time intervals should tend rapidly to zero." Please give example of such time interval.

- In the literature review, I could not pinpoint the knowledge gap. Furthermore, what was the investigators' research question and what was their hypotheses before listing the main objectives of this study. Also in the introduction, words like "an astonishing similarity" or "strikingly similar" can be a subjective interpretation of the results. I would like the authors to use quantitative comparisons instead.

- In the methods section, authors write "48 participants under the criteria that subjects’ recordings are present in all experimental conditions considered in this study.". Please elaborate more on the these experimental conditions.

- In the Methods section, please provide a diagram illustrating the experimental design.

- In the pre-statistical analysis phase, authors didn't give any justifications for channel selection, bandwidth, sampling frequency reduction, filter type...etc. Please elaborate on the rational behind your choice or was it all subjective? In other words, what happens to the results if I slightly use a different filter with a different bandwidth? will I get the same results?

- In Fig.1 , please add label and improve figure quality. In the methods section, please add the algorithms used to make the calculations.

- Fig. 2 is not that useful. Axes in a, b...f are not properly labeled. g and h are not clear. In addition, please increase text font size and figure quality. The same poor quality and lack of labeling and color bar applies to Fig. 3. Quality of Fig. 4 is not better.

Reviewer #2: The author of this study aimed at exploring the presence of stable interrelation patterns across different tasks, working memory and motor tasks, in MEG recordings from 48 clinically healthy participants. The main results reveal a pronounced stationary pattern across the scalp, similar to patterns observed in EEG signals under various conditions, highlighting the consistency of brain network interactions. Additionally, the study identifies the most effective EEG reference for analyzing brain functional networks through zero-lag cross-correlations, potentially contributing to complex systems theory near critical phase transitions. The study is innovative and potentially interesting for the PLOS ONE readers; however, I have some concerns that would like to be addressed before recommending it for publication, mostly related to statistical choices and justifications.

The statistical analysis described on MEG data incorporates several advanced techniques. However, it presents potential issues.

First, the statistical power sample. With a relatively small sample size (48 participants) I have concerns about the statistical power of the study, especially when probing complex interrelations among different physiological states. I understand the complexity of the participants selection in the Connectome dataset, but running a power analysis first, and include corrections for multiple comparisons later, can be helpful in support the power of your dataset.

Second, the use of the nonparametric Kolmogorov-Smirnov test to assess the expression magnitude of average cross-correlation structures is appropriate for data that may not follow a normal distribution. However, given the large number of comparisons (between conditions and subjects), the authors must address the issue of multiple comparisons and how they control for false discovery rates or family-wise error rates. Again, to ensure the robustness, the authors should conduct sensitivity analyses or bootstrap to assess the stability of their results across different sample subsets or parameter settings.

Third, comparing the observed with IAAFT surrogates is an innovative way to test for non-randomness in the data. Nevertheless, the authors should clearly explain the rationale behind using IAAFT surrogates, the generation process for these surrogates, and the interpretation of the comparison results. As for now, only the latter seems to be addressed, but this decision should be contextualized within the study's hypotheses and objectives, and not cited in discussion only.

Finally, cross-correlation and normalization. While normalizing the vectors to zero mean and unit variance before calculating Pearson correlation coefficients is a standard approach to measure topological similarity, the authors should discuss the potential limitations of this method, such as sensitivity to outliers or the assumption of linear relationships between variables.

As for the signal, I would expect to read a justification on the filtering and data segmentation process. 1Hz-30Hz fourth-order Butterworth band-pass filter between is common practice. However, the choice of filter settings (order and cutoff frequencies) can significantly affect the data, potentially introducing artifacts or altering signal characteristics. The authors should justify their choices based on the literature or empirical data. Additionally, the segmentation of data into 1.2-second intervals for analysis needs to be justified, particularly regarding how it might affect the analysis of neural responses that could extend beyond this interval.

Finally, the comparison of MEG with EEG using three different configurations (C1, C2, C3) is an intriguing attempt to integrate different modalities. This integration, however, introduces complexity in terms of data interpretation. The authors need to clarify in more detail how they account for the inherent differences in spatial resolution and signal characteristics between MEG and EEG in limitations.

Minor: there are some minor linguistics issues, that make some passages not easy to read. I suggest considering a syntax check and a careful rephasing of some passages, including the abstract.

To conclude, the paper presents an innovative and interesting idea, but needs justification of the main choices along with testing alternative data analysis and paying more attention to the syntax, so my recommendation is to provide a major review and a restructuring of the analysis.

6. PLOS authors have the option to publish the peer review history of their article (what does this mean?). If published, this will include your full peer review and any attached files.

Reviewer #1: No

Reviewer #2: No

---

## [Author Response · Author response to Decision Letter 0]

2 Apr 2024

Dear Editor, Dear Reviewer,

Below we comment on all the points you have raised.

Is done.

Is done.

4. Please expand the acronym “CoNaCyT” (as indicated in your financial disclosure) so that it states the name of your funders in full.

Is done in the actual version of the manuscript.

5. Please amend your manuscript to include your abstract after the title page.

Abstract is now included!

Comments to the Author

Reviewer #1: I have reviewed the manuscript entitled "Stationary Correlation Pattern in highly non-stationary MEG recordings of healthy subjects and its relation to former EEG studies". It is interesting and has some minor novelty but is not accepted until the authors address the following comments:

- The title includes "stationary correlation", but the manuscript did not include rigorous definitions nor implementation of stationarity tests.

We thank the referee for making us aware that some important pieces have been missing in the former version of the manuscript. We included in the introduction section the definition of stationarity and an ample description of the abstract terminology that stem from dynamical system theory but turns out to be necessary in the context of the paper. Furthermore, we included a description of different sources of non-stationarity, which may lead or not to the observation of the stationary pattern. By that we hope that also the relevance of our study gets more transparent. 

- Abstract is not well written. It is too convoluted and not clear. What do you mean by stable structures in "stable structures in otherwise highly nonstationary multivariate recordings"?

We deleted the first sentence of the abstract, which may indeed cause confusions. We also made corrections in the remaining part of the abstract. The phrase cited by the referee should remain clearer now by the detailed discussion of stationarity.

- Authors write that they analyzed Magnetoencephalographic recordings (MEG) of 48 clinically healthy subjects obtained by the Human Connectome Project (HCP). However, in the literature review, many of the examples were based on signals with neurological disorders. Please explain the level of accuracy in this literature review?

Many of the studies cited and described in the introduction are not concerned with pathologies. Two contributions of Dra. Mari Corsi used EEG recordings of clinically healthy women, The Contribution of Olguín et al. presents the results obtained for 10 healthy subjects during the second night sleep, 11 elderly and 10 young subjects at rest during open and closed eyes, beside of the EEGs that contain the peri-ictal transition of 9 epilepsy patients. And the paper published by Ramos-Loyo presents the analysis of three groups of healthy children between 6 and 10 years of age. Thus, in fact most of the subjects reported in the manuscript are clinically healthy. However, the fact we observe a high similarity between e.g. sleep EEGs or recordings of healthy subjects with open or closed eyes and those of epilepsy patients before, during or after an epileptic crisis underlines the robustness of the observation reported in the manuscript.

- In the Introduction, authors write "over longer time intervals should tend rapidly to zero." Please give example of such time interval.

Citing the paper of Kramer et al, we specified already a typical time scale, namely 100 seconds for intracranial recordings. We further stated in the text that “Note, this is true for averages over considerably long periods of at least one or two minutes. On shorter time scales, one may observe notable fluctuations (see e.g., Fig. 6 of Olguín-Rodriguez et al. 2018).” This provides already a clear estimate of the corresponding time scales. In the current version of the manuscript, however, we present additional results in the form of a figure that allows us to estimate such a time scale. Hereby we arrive at an estimate of approx. 150 seconds for extracranial data.

- In the literature review, I could not pinpoint the knowledge gap. Furthermore, what was the investigators' research question and what was their hypotheses before listing the main objectives of this study. Also in the introduction, words like "an astonishing similarity" or "strikingly similar" can be a subjective interpretation of the results. I would like the authors to use quantitative comparisons instead.

In the paragraphs where we included a definition and discussion about stationarity, we also discuss different sources of nonstationary behavior. The comparison with MEG recordings provides the possibility to distinguish between the different scenarios. Thus, we hope that the “knowledge gap” gets clearer in the actual version. We also articulate a clear hypothesis in the actual version of the manuscript.

Concerning the terminology like “astonishing similarity” we exchanged these expressions by quantitative comparisons. In all cases the distribution of Pearson coefficients for the comparison of two average cross-correlation matrices are centered above 0.75 and occasionally (e.g. sleep recordings) all coefficients take values above 0.9.

- In the methods section, authors write "48 participants under the criteria that subjects’ recordings are present in all experimental conditions considered in this study.". Please elaborate more on the these experimental conditions.

We included a more detailed description of the experiment and the subject selection. However, we like to emphasize that the specific cognitive or motor task of the MEG experiment is of minor importance for our study. The main point is, that although the MEG-recordings as well as the different EEG measurements have been undertaken in quite different experimental setups and that the subjects were in quite different physiological conditions, we observe nevertheless that (i) the MEG recordings show on the average a pronounced stable cross-correlation pattern that is (ii) highly similar to the SCP observed for the EEGs. This result is robust in the sense that we probe three different schemes to reduce the number of MEG detectors simulating the EEG 10-20 system, obtaining quantitatively similar results.

- In the Methods section, please provide a diagram illustrating the experimental design.

We included a diagram of the experimental design, namely the pre-processing pipeline and the statistical analysis, in the actual version of the manuscript.

- In the pre-statistical analysis phase, authors didn't give any justifications for channel selection, bandwidth, sampling frequency reduction, filter type...etc. Please elaborate on the rational behind your choice or was it all subjective? In other words, what happens to the results if I slightly use a different filter with a different bandwidth? will I get the same results?

In the actual version we included information about filter borders, filter type, arguments in favor of our choice of the filter type, the rational behind the down sampling etc. MEG channel selection was done in three different schemes as described in the text and listed in a separate table to mimic the international 10-20 system of EEG recordings. It turns out that the results are quantitatively almost independent of the chosen reduction scheme, which emphasizes the robustness of our results.

Furthermore, we like to emphasize that one observes the stationary pattern also in different frequency bands. Please see below cross-correlation matrices averaged over a whole night recording of signals filtered in different frequency bands.

We did not include a corresponding comment or figure in the manuscript because we reserve this material for a forthcoming paper.

- In Fig.1 , please add label and improve figure quality. In the methods section, please add the algorithms used to make the calculations.

Labels are added in the actual version of the manuscript and the quality has been improved. Actually, the quality of the figures uploaded by us have a much better quality as the compiled version of the paper.

- Fig. 2 is not that useful. Axes in a, b...f are not properly labeled. g and h are not clear. In addition, please increase text font size and figure quality. The same poor quality and lack of labeling and color bar applies to Fig. 3. Quality of Fig. 4 is not better.

We added labels, increased text font and improved the quality of the figures.

Reviewer #2: The author of this study aimed at exploring the presence of stable interrelation patterns across different tasks, working memory and motor tasks, in MEG recordings from 48 clinically healthy participants. The main results reveal a pronounced stationary pattern across the scalp, similar to patterns observed in EEG signals under various conditions, highlighting the consistency of brain network interactions. Additionally, the study identifies the most effective EEG reference for analyzing brain functional networks through zero-lag cross-correlations, potentially contributing to complex systems theory near critical phase transitions. The study is innovative and potentially interesting for the PLOS ONE readers; however, I have some concerns that would like to be addressed before recommending it for publication, mostly related to statistical choices and justifications.

We thank the referee for the overall positive opinion about our work.

The statistical analysis described on MEG data incorporates several advanced techniques. However, it presents potential issues.

First, the statistical power sample. With a relatively small sample size (48 participants) I have concerns about the statistical power of the study, especially when probing complex interrelations among different physiological states. I understand the complexity of the participants selection in the Connectome dataset, but running a power analysis first, and include corrections for multiple comparisons later, can be helpful in support the power of your dataset.

We thank the referee to make us aware about the important missing information that supports and validates the presented results. The oost-hoc power two tails test for Pearson correlation coefficients revealed a power value of 0.994 with a p-value of 0.001 considering a sample of 48 subjects and a correlation value of 0.7, a smaller value that the observed in the correlation analysis. Additionally, Bonferroni post-hoc tests for multiple comparisons were also estimated.

Second, the use of the nonparametric Kolmogorov-Smirnov test to assess the expression magnitude of average cross-correlation structures is appropriate for data that may not follow a normal distribution. However, given the large number of comparisons (between conditions and subjects), the authors must address the issue of multiple comparisons and how they control for false discovery rates or family-wise error rates. Again, to ensure the robustness, the authors should conduct sensitivity analyses or bootstrap to assess the stability of their results across different sample subsets or parameter settings.

Please note, the Kolmogorov Smirnov test is applied for the comparison with the surrogate data, which is of course a kind of boot strapping. At this point it is important to account for data that represent adequately the null hypothesis, that the results obtained for the empirical recordings are not due to the presence of genuine cross-correlations, but they are exclusively a statistical effect caused by so called random correlations. Thus, we must conserve this part, which is done by considering IAAFT surrogates. Please note further that the obtained p-value is below 10-6, which implies that even with a Bonferroni correction we are far below the 1% significance level. However, we thank the referee for this remark, given that it makes us aware of a deficient redaction. Consequently, we have included several comments on this topic with appropriate references in the main text.

Third, comparing the observed with IAAFT surrogates is an innovative way to test for non-randomness in the data. Nevertheless, the authors should clearly explain the rationale behind using IAAFT surrogates, the generation process for these surrogates, and the interpretation of the comparison results. As for now, only the latter seems to be addressed, but this decision should be contextualized within the study's hypotheses and objectives, and not cited in discussion only.

We hope that the additional comments on this topic, added in the method section together with the relevant citations, will make this aspect transparent in the current version of the text.

Finally, cross-correlation and normalization. While normalizing the vectors to zero mean and unit variance before calculating Pearson correlation coefficients is a standard approach to measure topological similarity, the authors should discuss the potential limitations of this method, such as sensitivity to outliers or the assumption of linear relationships between variables.

We added a paragraph explaining why we are focusing on linear cross-correlations. We further underline that we do not claim that nonlinear interrelations are less important, but here we aim to compare quantitatively our results with those obtained in former studies where linear interrelations have been used (beside of the fact that it seems that surrogate corrected mutual information does not lead to any signature of a stationary pattern of non-linear interrelations). 

We would also like to clarify that normalization does not change the sensitivity to outliers within the data segment used to estimate the linear correlations. The standard procedure of the normalization moves the average to zero and scales the standard deviation to unity. Covariance and correlation are equally vulnerable to outliers.

As for the signal, I would expect to read a justification on the filtering and data segmentation process. 1Hz-30Hz fourth-order Butterworth band-pass filter between is common practice. However, the choice of filter settings (order and cutoff frequencies) can significantly affect the data, potentially introducing artifacts or altering signal characteristics. The authors should justify their choices based on the literature or empirical data. Additionally, the segmentation of data into 1.2-second intervals for analysis needs to be justified, particularly regarding how it might affect the analysis of neural responses that could extend beyond this interval.

We included a paragraph justifying the choice of the filter borders, and the filter type used by us, comment on the down sampling and other technical details of the data pre-processing, including several citations relevant in this context. We also added an additional comment concerning the choice of 1.2 second segments.

Finally, the comparison of MEG with EEG using three different configurations (C1, C2, C3) is an intriguing attempt to integrate different modalities. This integration, however, introduces complexity in terms of data interpretation. The authors need to clarify in more detail how they account for the inherent differences in spatial resolution and signal characteristics between MEG and EEG in limitations.

We added a corresponding comment in the method section where we present the three schemes of reducing the number of MEG channels with the aim to simulate the international 10-20 system of EEG recordings.

Minor: there are some minor linguistics issues, that make some passages not easy to read. I suggest considering a syntax check and a careful rephasing of some passages, including the abstract.

We apologize that some passages were not easy to read. We have rephrased and extended some explanations in order to me more explicit in the actual version of the manuscript.

To conclude, the paper presents an innovative and interesting idea, but needs justification of the main choices along with testing alternative data analysis and paying more attention to the syntax, so my recommendation is to provide a major review and a restructuring of the analysis.

We appreciate this affirmative comment! We have reconstructed the analysis including two new figures and a power anal

---

## [Decision Letter · Decision Letter 1]

7 May 2024

PONE-D-23-40266R1Stationary Correlation Pattern in highly non-stationary MEG recordings of healthy subjects and its relation to former EEG studiesPLOS ONE

Dear Dr. Müller,

Thank you for submitting your manuscript to PLOS ONE. After careful consideration, we feel that it has merit but does not fully meet PLOS ONE’s publication criteria as it currently stands. Therefore, we invite you to submit a revised version of the manuscript that addresses the points raised during the review process.

Specifically in the light of reviewers' comments and feedback: minor review is recommended, please.

We look forward to receiving your revised manuscript.

Kind regards,

Umer Asgher, PhD

Academic Editor

PLOS ONE

Journal Requirements:

Reviewers' comments:

Reviewer's Responses to Questions

**Comments to the Author**

1. If the authors have adequately addressed your comments raised in a previous round of review and you feel that this manuscript is now acceptable for publication, you may indicate that here to bypass the “Comments to the Author” section, enter your conflict of interest statement in the “Confidential to Editor” section, and submit your "Accept" recommendation.

Reviewer #1: (No Response)

Reviewer #2: All comments have been addressed

2. Is the manuscript technically sound, and do the data support the conclusions?

Reviewer #1: Yes

Reviewer #2: Yes

3. Has the statistical analysis been performed appropriately and rigorously? 

Reviewer #1: Yes

Reviewer #2: Yes

4. Have the authors made all data underlying the findings in their manuscript fully available?

Reviewer #1: Yes

Reviewer #2: Yes

5. Is the manuscript presented in an intelligible fashion and written in standard English?

Reviewer #1: Yes

Reviewer #2: Yes

6. Review Comments to the Author

Reviewer #1: The authors have addressed most of my comments. I still thing the Abstract is no of high quality. Authors did not address my comment on the abstract and still using words like strikingly similar without quantities although in their response they write that they have done that. They did not address my comment on Fig. 1. The figure added to illustrate the experimental design is not clear. There are some careless English mistakes like "hypothesise". Authors need to improve the English language.

Reviewer #2: the authors have provided satisfactory responses to all of my concerns, made explanations when necessary and changed the text accordingly.

7. PLOS authors have the option to publish the peer review history of their article (what does this mean?). If published, this will include your full peer review and any attached files.

Reviewer #1: No

Reviewer #2: No

---

## [Author Response · Author response to Decision Letter 1]

23 May 2024

The reviewer claims that: “The authors have addressed most of my comments. I still thing the Abstract is no of high quality. Authors did not address my comment on the abstract and still using words like strikingly similar without quantities although in their response they write that they have done that. They did not address my comment on Fig. 1. The figure added to illustrate the experimental design is not clear. There are some careless English mistakes like "hypothesise". Authors need to improve the English language.”

We changed Figure 1 and Figure 2, hoping that the new versions of both figures are now satisfying. However, we would like to remark that it is certainly no sufficient to state that a Figure is not clear, but one should specify possible deficiencies. Furthermore, in contrast to the statement of the reviewer we added labels to Fig. 1 and explained the algorithm in the main text. However, here we propose a new design of Figure 1.

We changed the abstract, actually we rewrote it completely. However, we think that using a term like “strikingly” in this part of a manuscript is by no means a deficient redaction and we cannot see any rational behind the claim of the reviewer. We are still of the opinion that the use of this term was justified, especially since we quantitatively demonstrate in the main text how highly (strikingly) similar the correlation structures are.

Finally, using the term “hypothesise” or “hypothesize” is a question of using British or the American way of English writing. But it is certainly not a mistake, as the reviewer claims. Besides, we recommend the reviewer being more careful in the redaction of his report (see e.g. “I still thing the Abstract is no of high quality”), especially when he/she criticizes the English of others. Being consistent within the whole text we did not change this term.

---

## [Decision Letter · Decision Letter 2]

4 Jul 2024

Stationary Correlation Pattern in highly non-stationary MEG recordings of healthy subjects and its relation to former EEG studies

PONE-D-23-40266R2

Dear Dr. Müller,

We’re pleased to inform you that your manuscript has been judged scientifically suitable for publication and will be formally accepted for publication once it meets all outstanding technical requirements.

Kind regards,

Umer Asgher, PhD

Academic Editor

PLOS ONE

Additional Editor Comments (optional):

Reviewers' comments:

Reviewer's Responses to Questions

**Comments to the Author**

1. If the authors have adequately addressed your comments raised in a previous round of review and you feel that this manuscript is now acceptable for publication, you may indicate that here to bypass the “Comments to the Author” section, enter your conflict of interest statement in the “Confidential to Editor” section, and submit your "Accept" recommendation.

Reviewer #1: All comments have been addressed

2. Is the manuscript technically sound, and do the data support the conclusions?

Reviewer #1: Yes

3. Has the statistical analysis been performed appropriately and rigorously? 

Reviewer #1: Yes

4. Have the authors made all data underlying the findings in their manuscript fully available?

Reviewer #1: Yes

5. Is the manuscript presented in an intelligible fashion and written in standard English?

Reviewer #1: Yes

6. Review Comments to the Author

Reviewer #1: I have reviewed the revised manuscript and see that the authors have addressed my comments. Thank you! I would like to offer one suggestion: please avoid taking a negative view of reviewer comments. The intention of the critique is to enhance the quality of the manuscript, not to criticize the authors or their work.

7. PLOS authors have the option to publish the peer review history of their article (what does this mean?). If published, this will include your full peer review and any attached files.

Reviewer #1: No

---

## [Editor Report · Acceptance letter]

21 Aug 2024

PONE-D-23-40266R2 

PLOS ONE

Dear Dr. Müller, 

I'm pleased to inform you that your manuscript has been deemed suitable for publication in PLOS ONE. Congratulations! Your manuscript is now being handed over to our production team.

Kind regards, 

on behalf of

Dr. Umer Asgher 

Academic Editor

PLOS ONE